# Definitions matter: Heterogeneity of COVID-19 disease severity criteria and incomplete reporting compromise meta-analysis

**Philippe J. Guérin**[1,2]*, **Alistair R. D. McLean**[1,2], **Sumayyah Rashan**[1,2], **AbdulAzeez Lawal**[1,2], **James A. Watson**[2,3], **Nathalie Strub-Wourgaft**[4], **Nicholas J. White**[2,3]

**1** Infectious Diseases Data Observatory (IDDO), Oxford, United Kingdom, **2** Nuffield Department of Medicine, Centre for Tropical Medicine & Global Health, University of Oxford, Oxford, United Kingdom, **3** Faculty of Tropical Medicine, Mahidol Oxford Tropical Medicine Research Unit (MORU), Mahidol University, Bangkok, Thailand, **4** Drugs for Neglected Diseases Initiative, Geneva, Switzerland

* philippe.guerin@iddo.org

**Data Availability Statement:** All the data used in this study are publicly available and properly cited. The analysis database if freely available "Guerin, P.

## Abstract

Therapeutic efficacy in COVID-19 is dependent upon disease severity (treatment effect heterogeneity). Unfortunately, definitions of severity vary widely. This compromises the meta-analysis of randomised controlled trials (RCTs) and the therapeutic guidelines derived from them. The World Health Organisation 'living' guidelines for the treatment of COVID-19 are based on a network meta-analysis (NMA) of published RCTs. We reviewed the 81 studies included in the WHO COVID-19 living NMA and compared their severity classifications with the severity classifications employed by the international COVID-NMA initiative. The two were concordant in only 35% (24/68) of trials. Of the RCTs evaluated, 69% (55/77) were considered by the WHO group to include patients with a range of severities (12 mild-moderate; 3 mild-severe; 18 mild-critical; 5 moderate-severe; 8 moderate-critical; 10 severe-critical), but the distribution of disease severities within these groups usually could not be determined, and data on the duration of illness and/or oxygen saturation values were often missing. Where severity classifications were clear there was substantial overlap in mortality across trials in different severity strata. This imprecision in severity assessment compromises the validity of some therapeutic recommendations; notably extrapolation of "lack of therapeutic benefit" shown in hospitalised severely ill patients on respiratory support to ambulant mildly ill patients is not warranted. Both harmonised unambiguous definitions of severity and individual patient data (IPD) meta-analyses are needed to guide and improve therapeutic recommendations in COVID-19. Achieving this goal will require improved coordination of the main stakeholders developing treatment guidelines and medicine regulatory agencies. Open science, including prompt data sharing, should become the standard to allow IPD meta-analyses.

(2022). Analysis Database for "Definitions matter: heterogeneity of COVID-19 disease severity criteria and incomplete reporting compromise meta-analysis", Harvard Dataverse. https://doi.org/10.7910/DVN/YORCZN.

**Funding:** This work is partly supported by the COVID-19 Clinical Research Coalition, funded by the Federal Ministry of Education and Research (bundesministerium für bildung und forschung) through KfW, Germany (ref: 2020 62 156) to PG, and the Republic and the canton de genève, International Solidarity Service, Switzerland (ref convention 2020) to PG. The funders had no role in study design, data collection and analysis, decision to publish, or preparation of the manuscript.

**Competing interests:** I have read the journal's policy and the authors of this manuscript have the following competing interests: All authors have completed the ICMJE uniform disclosure form at www.icmje.org/coi_disclosure.pdf and declare: JAW and NJW are funded by the Wellcome Trust (ref: 220211/Z/20/Z and 093956/Z/10/C)

## Introduction

Viral burdens in COVID-19 infection peak early, around the time of symptom onset, and then decline. In the minority of patients requiring hospitalisation and respiratory support, inflammatory processes dominate [1,2]. As a result, drugs used to treat COVID-19 may have varying efficacy depending on where the patient is in the disease course when the medicines are administered. Under this simple paradigm antiviral drugs would be expected to be most beneficial when administered as early as possible in the evolution of the individual infection, and less likely to benefit once inflammatory processes dominate later in the disease [3]. At this late stage [4], immune modulating and anti-inflammatory drugs are of proven benefit. Conversely, immune suppression could be harmful early in the infection by attenuating an effective early immune response to viral replication. The results of the seminal RECOVERY trial on dexamethasone provide some support for this paradigm, with evidence of treatment effect heterogeneity according to patient severity at randomisation, a proxy measurement of disease progression [5]. Low dose dexamethasone (6mg/day) resulted in lower 28-day mortality among hospitalised patients who were receiving either invasive mechanical ventilation at randomization (RR = 0.64; 95% CI, 0.51–0.81) or oxygen alone at randomization (RR = 0.82; 95% CI, 0.72–0.94). In contrast, among those receiving no respiratory support at randomisation, the mortality figures were in the direction of harm (RR = 1.19; 95% CI, 0.92–1.55, p<0.001 for trend across three groups). The WHO guidelines for the treatment of COVID-19 acknowledged this interaction by providing separate recommendations stratified by disease severity for corticosteroids, recommending their use in severe and critical patients whilst conditionally recommending against their use in non-severe patients [6]. Conversely, monoclonal antibodies targeting the virus spike proteins have shown benefit for mild to moderately ill patients, whilst their efficacy was not demonstrated in more severely ill cases [7,8].

In the absence of precise knowledge of when a patient was infected, clinicians use history and disease severity at presentation to assess the appropriate clinical management. Disease severity is a continuous spectrum but, as is common for many potentially severe infectious diseases, researchers have partitioned COVID-19 severity into discrete categories of mild, moderate, severe and critical. This is useful for triage and, if these definitions were consistent, would also be useful for comparison of clinical and epidemiological observations, investigations and trials–as in the COVID-19 Network Meta-Analysis (NMA) underpinning the WHO guidelines. Most groups use the oxygen saturation level ($SpO_2$) at rest in ambient air; respiratory rate (breaths per minute) and–when available–the ratio of arterial oxygen partial pressure (mmHg) to fractional inspired oxygen ($PaO_2/FiO_2$) in their severity criteria. However, the thresholds used vary substantially. The US National Institutes of Health (NIH) and the World Health Organization (WHO) have proposed different sets of criteria to categorise patients by severity (Fig 1 and S1 Table). Most notably these criteria differ on the saturation of oxygen ($SpO_2$) threshold to define a severe case (the NIH considers an individual with <94% $SpO_2$ to be a severe case, whilst the WHO requires <90% $SpO_2$). The living network meta-analysis underpinning the WHO treatment guidelines, referred hereafter as "WHO–COVID19 Living Network Meta-Analysis" [6,9] and the international COVID-NMA Initiative [10,11] both use different severity definitions ((Fig 1 and S1 Table).

When the efficacy of a drug varies depending upon disease severity at time of treatment, as in COVID-19, harmonised definitions of severity are essential to provide severity specific estimates of drug efficacy. So, if definitions of patient severity vary across published reports, stratified efficacy estimates are likely to be compromised. Severity specific estimates of drug efficacy are compromised further if definitions based on clinical observations are not accompanied by pertinent data on date of symptoms onset and/or oxygen saturation values. We sought to determine the feasibility of retrospectively classifying the disease severity of patients included

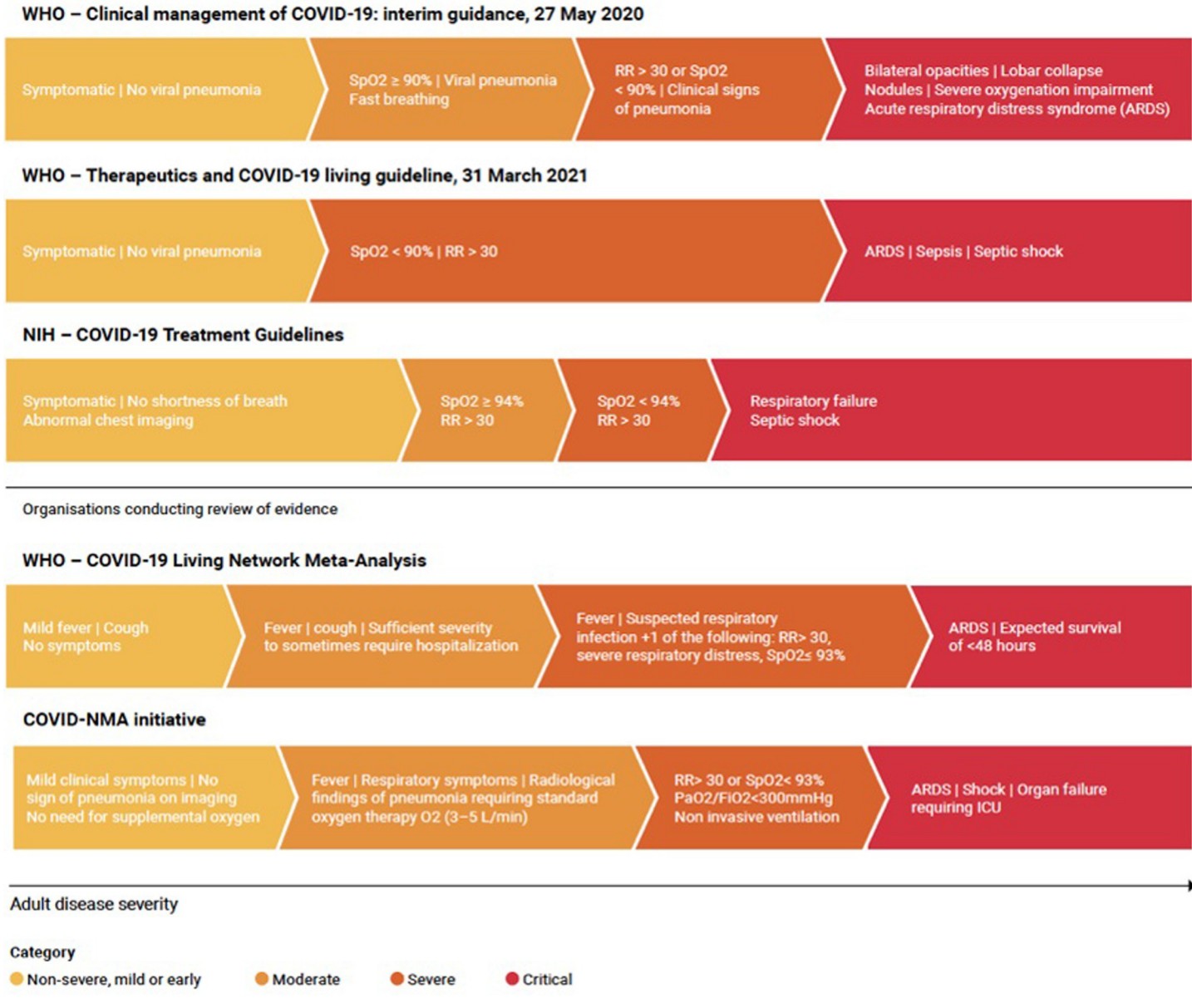

**Fig 1. Severity definitions from WHO, US NIH guidelines and organisations conducting review of evidence i.e. the WHO—COVID-19 Living Network Meta-Analysis and the COVID-NMA imitative (RR: Respiratory rate; SpO2: Blood Oxygen Saturation; PaO2/FiO2: Ratio of arterial oxygen partial pressure (PaO2 in mmHg) to fractional inspired oxygen (FiO2 expressed as a fraction)).**

in the COVID-19 clinical trials upon which the current WHO living therapeutic guidelines are based. This was obtained from published information by extracting severity components from the clinical trial publications included in the living network meta-analysis.

## Materials and methods

### Identification of studies

We evaluated the 85 trials that met the inclusion criteria for the WHO—COVID-19 Living Network Meta-Analysis for drug treatments for COVID-19 (update 2, published 17[th] of

December 2020). From these 85 trials, we could extract information from 81 (S2 Table). The full texts of the remaining four were inaccessible to us by 1st May 2021.

## Categorical indicators of disease severity

We extracted information on the following variables and categorical indicators of COVID-19 severity at baseline: outpatients; patients with pneumonia; patients receiving oxygen therapy; inpatients; ICU patients. We extracted whether a) the results section (including the baseline table) indicated that any participants had any of the indicators listed above; b) if any of the indicators were listed as an exclusion criterion; c) if any of the indicators were listed as a necessary inclusion criterion; and d) if any of the indicators were listed as a sufficient but not necessary inclusion criterion. We then determined whether in the study all, some or no participants had any of the indicators, or if it was not known if any participants had any of the indicators or not.

## Continuous indicators of severity

We extracted information on the following continuous indicators of COVID-19 severity at baseline: days since symptom onset; oxygen saturation level ($SpO_2$) at rest in ambient air; respiratory rate (breaths per minute); the ratio of arterial oxygen partial pressure (mm Hg) to fractional inspired oxygen ($PaO_2/FiO_2$). For all continuous indicators we extracted the following measures where available: median; lower quartile; upper quartile; mean; standard deviation; observed minimum in study population; observed maximum in study population; minimum inclusion criteria where the inclusion criteria were sufficient but not necessary; maximum inclusion criteria where the inclusion criteria were sufficient but not necessary; minimum inclusion criteria where the inclusion criteria were necessary; maximum inclusion criteria where the inclusion criteria were necessary; proportions and ranges of categorisations. Where the observed measures were presented separately by arm we extrapolated overall study information where possible (in the case of the minimum and maximum measures for continuous indicators). Where study level measures were not identifiable, measures reported for the arm with the larger sample size were used. In addition, we classified whether the observed measures we extracted are representative of the entire study population or a single study arm.

## Network Meta-Analysis (NMA) classifications

We extracted the mean age, mortality and possible severity range of patients listed online by the WHO–COVID-19 Living Network Meta-Analysis group [6,9] which provide evidence to WHO (https://www.covid19lnma.com/drug-treatments-study-level). Where the WHO–COVID-19 Living Network Meta-Analysis group listed a study arm as having "1 or 2" deaths in it (Zhao 2020) we did not include this arm in our calculation of mortality. We also extracted the severity classifications according to the COVID-NMA Initiative [10,11] (https://covid-nma.com/living_data/). We used the data available at these sites as on the 4th of March 2021.

The severity of two studies [12,13] included in update 2 of the WHO—COVID-19 Living Network Meta-Analysis group site were not present on the website so these were extracted from the original manuscript. Where severity was captured as not reported (NR) by the WHO—COVID-19 Living Network Meta-Analysis group we considered that patients with this severity may have been present in the trial and therefore considered this as part of the possible severity range of patients.

## Data analysis

Data were summarised using descriptive statistics including percentages and counts where appropriate. Figures were generated using Stata 17.0 (College Station, TX, USA).

**Ethical approval.**  As all the data were anonymised, available in the public domain and aggregated without any personal information, ethics approval was deemed unnecessary.

## Results

### Severity classification according to NMA groups

Of the 81 studies (S2 Table), 24 (30%) studies included patients from one severity category only in the WHO—COVID-19 Living Network Meta-Analysis (5 studies mild patients only; 8 moderate only; 7 severe only; 4 critical only) ((Fig 2). The majority (57/81, 70%) of studies included patients with a range of clinical severities (12 mild-moderate; 3 mild-severe; 19 mild-critical; 5 moderate-severe; 8 moderate-critical; 10 severe-critical). The severity classification used by the COVID-NMA initiative usually did not align with that used by the WHO—COVID-19 Living Network Meta-Analysis group. Of 70 studies that had been classified by both sources (The COVID-NMA initiative did not list four trials, six trials had "unclear" severity and one trial was unclassified), only 26 (37%) (of studies were classified as having the same range of severity by both groups (S3 Table). In studies with a single patient severity category, as expected, mortality rates stratified roughly according to severity. However, in studies with multiple severity strata, there was considerable overlap in mortality across the different severity definitions with variation not explained by mean patient age (Fig 3).

### Categorical indicators of severity

We extracted information on whether patients included in the trials were inpatients; outpatients; hospitalised in ICU; diagnosed with pneumonia; and receiving oxygen therapy. The reporting of whether all, some or no patients had each categorical indicator of severity are shown in Fig 4. In the large majority of studies 65/81 (80%) all patients were inpatients, while in 7 (9%) all patients were outpatients, 5 (6%) contained some inpatients but it was unclear if any outpatients were included, while in the remaining 4 (5%) studies it could not be ascertained if outpatients or inpatients were included. In the majority (54/65, 83%) of non-outpatient studies it was unclear if any of the patients were in ICU at baseline. Four (5%) studies reported all patient were in ICU; six (7%) reported some; and 17 (21%) reported none. In 33/81 (41%) studies all patients had pneumonia, with some patients with pneumonia reported in 19 (23%) studies. It was unknown in the remaining 29 (36%) studies. In the majority (47/81, 58%) of studies it was unclear if any of the patients were on oxygen therapy at baseline. Only one (1%) study indicated that no patients were on oxygen therapy, one (1%) study indicated that all patients were on oxygen therapy, and the remaining 32/79 (40%) reported that some patients were receiving oxygen.

### Continuous indicators

Of the 81 studies, the majority, 54 (66%) provided information on the intervals from symptom onset ((Fig 5); 37 studies (46%) presented days since symptom onset as quartiles of the patient population; 13 studies gave means and standard deviations; three studies reported the minimum eligible in terms of day of onset for inclusion; 13 studies reported the maximum time eligible; two studies reported the observed minimum and maximum; and one study reported proportions of individuals falling into categorisations. Of the studies that reported a minimum number of days since symptom onset threshold to be eligible for inclusion, two reported 3 days and one reported 7 days. Of the studies that reported a maximum number of days since symptom onset threshold to be eligible for inclusion, three reported 4 days, one reported 6

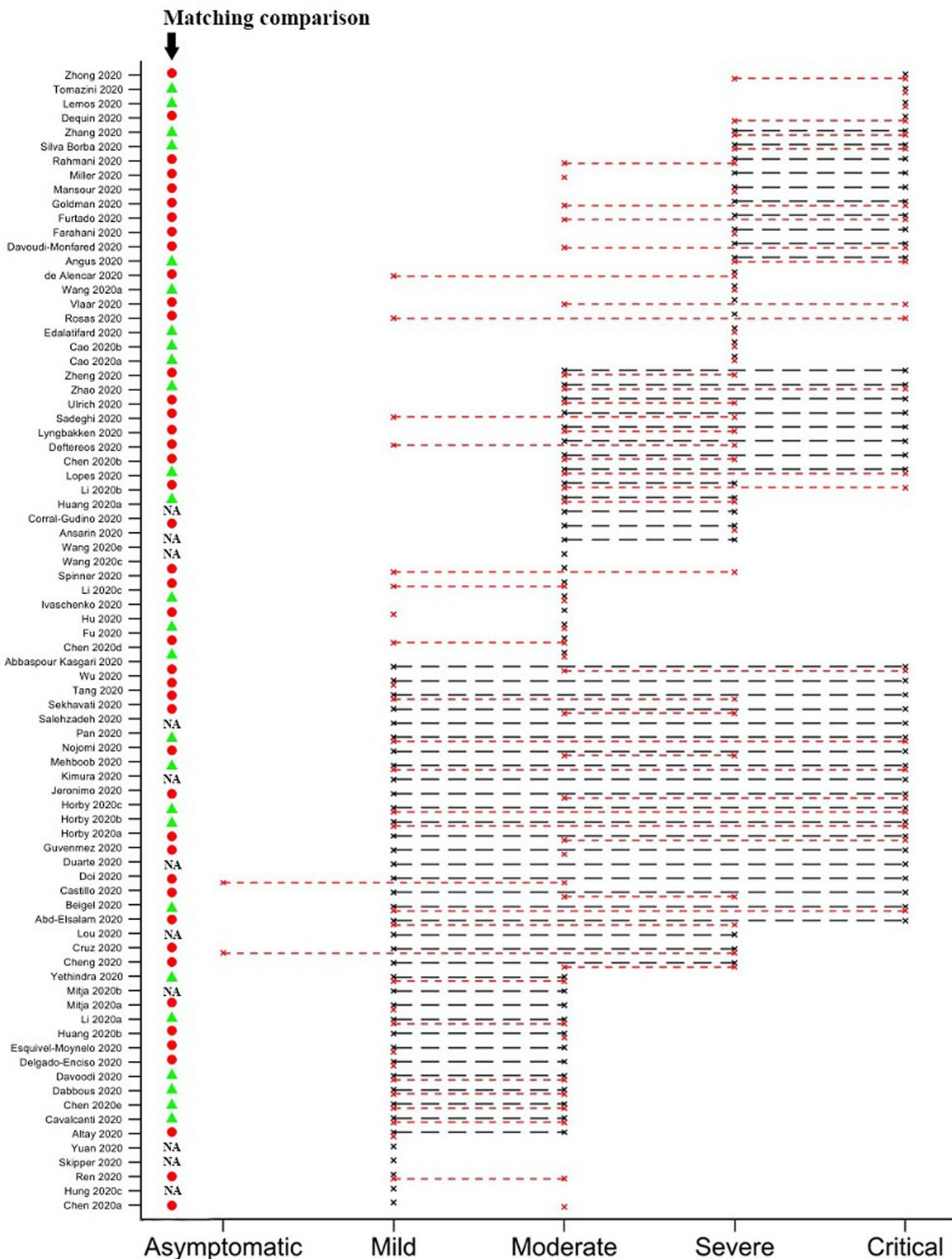

**Fig 2. Possible range of patient severity in studies as classified by WHO- COVID-19 Living Network Meta-Analysis group (black circles) and COVID-NMA initiative (red squares).** Matched severity definiton between the two groups (green triangle), unmatched (red circle).

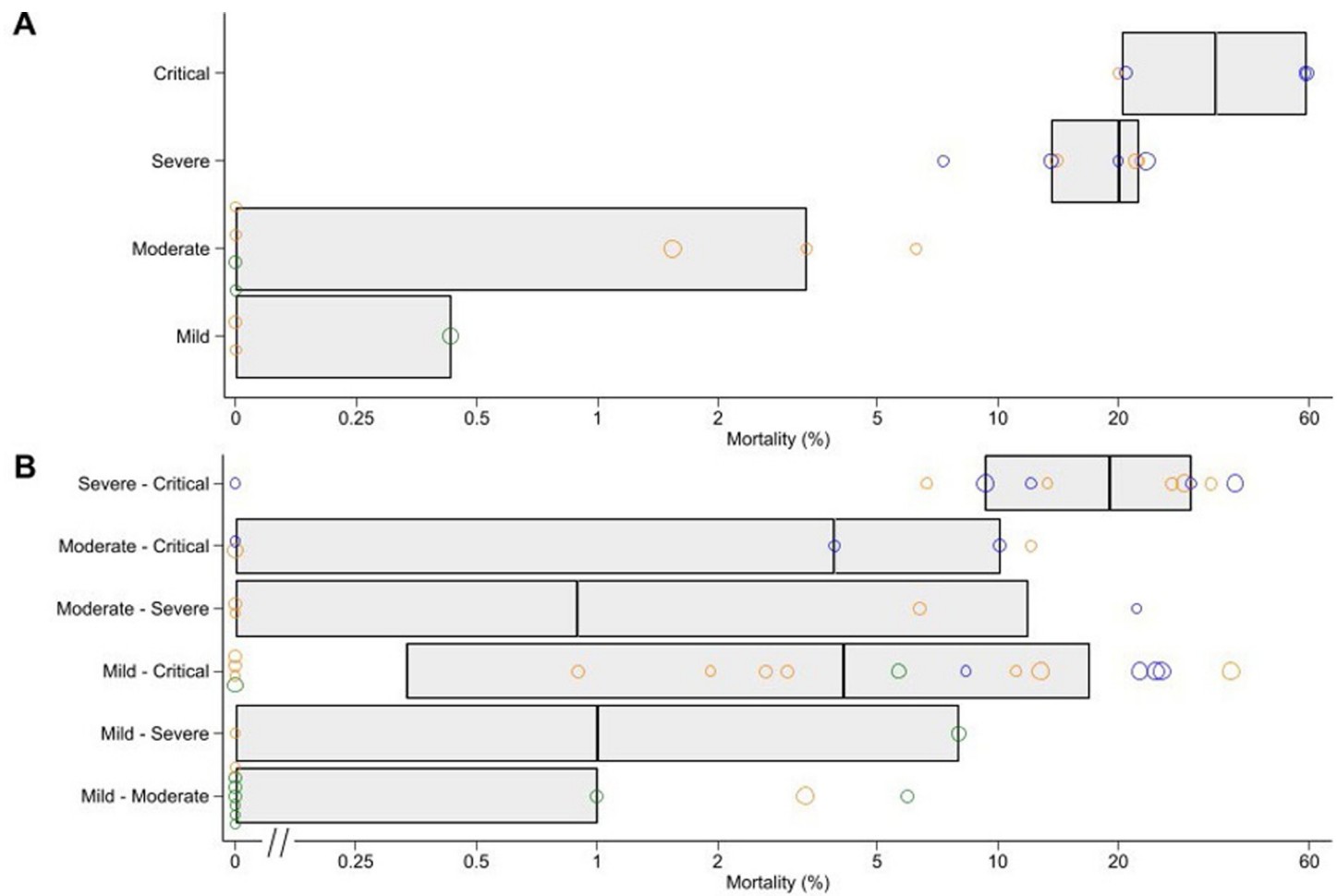

**Fig 3.** Trial mortality among trials with (A) only a single stratum of patient severity and (B) a range of possible patient severity as reported by https://www.covid19lnma. com/drug-treatments-study-level. Green circles are from trials with mean patient age<50 years; orange for trials with mean patient age 50–60 years; blue for trials with mean patient age >60 years. Boxes denote quartiles. Circle width corresponds to number of participants analysed (<75 participants; 75–150 participants; 150–300 participants; > = 300 participants from smallest to largest).

days, two reported 8 days, one reported 10 days, three reported 12 days, one reported 13 days and two reported 14 days.

Information on $SpO_2$ was provided in 44 studies (54%). Eleven studies presented $SpO_2$ quartiles of the patient population; 12 studies gave means and standard deviations; 14 studies reported the minimum eligible for inclusion and three studies reported the maximum eligible for inclusion; 18 studies reported a maximum threshold of $SpO_2$ that was sufficient but not necessary for inclusion; no studies reported the observed minimum and maximum; and two studies reported proportions of individuals falling within various $SpO_2$ ranges. There was substantial heterogeneity in $SpO_2$ thresholds. Of the studies that reported a minimum $SpO_2$ threshold required for inclusion three gave a threshold of 95%, six gave 94%, three gave 93%, one gave 90% and one gave 75%. Of the studies that reported a maximum $SpO_2$ threshold to be eligible for inclusion two gave 94% and one gave 90%.

Only 41 of 81 (51%) studies reported information on the respiratory rates (RR) of their patients. There were 16 studies which presented RR quartiles of the patient population; 11 studies gave means and standard deviations; three studies reported the minimum eligible for

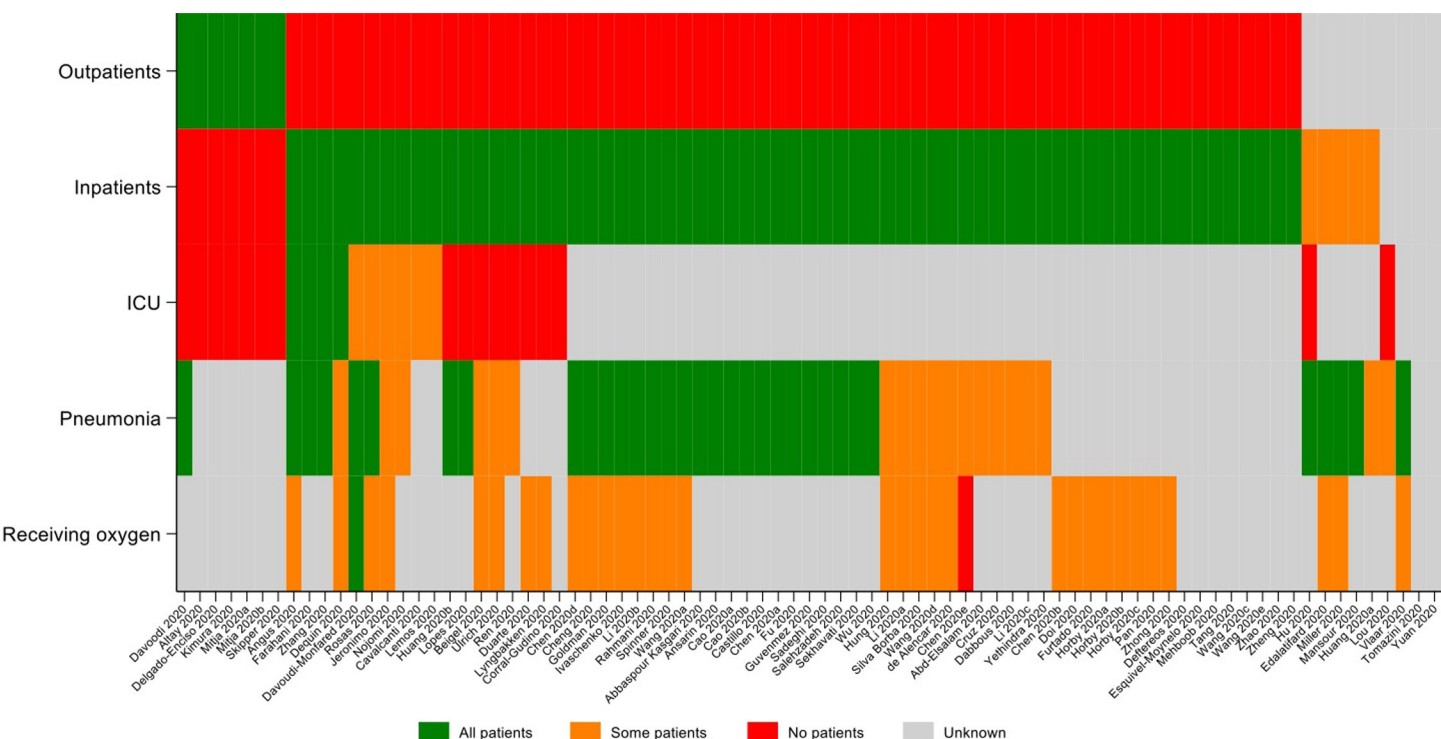

**Fig 4. Heatplot of studies and whether any patients in the study were outpatients; inpatients; ICU patients; patients with pneumonia; patients receiving oxygen at baseline.**

inclusion and nine studies reported the maximum eligible for inclusion; nine studies reported a maximum threshold of RR that was sufficient, but not necessary, for inclusion and one study reported a minimum threshold of RR that was sufficient, but not necessary, for inclusion; seven studies reported proportions of individuals falling into categorisations. Of the studies that reported a minimum RR threshold to be eligible for inclusion there was one each for thresholds of 19, 25 and 30 breaths/minute. Of the studies that reported a maximum threshold to be eligible for inclusion one gave 23, four gave 29, three gave 30 and one gave 35 breaths/minute.

Only 29 of 81 (36%) studies reported information on $PaO_2/FiO_2$ ratios. Four studies presented $PaO_2/FiO_2$ quartiles of the patient population; five studies gave means and standard deviations; eight studies reported the minimum eligible for inclusion and five studies reported the maximum eligible for inclusion; thirteen studies reported a maximum threshold of $SpO_2$ that was sufficient but not necessary for inclusion. Of the studies that reported a minimum threshold for inclusion, one gave a $PaO_2/FiO_2 = 76$, one gave 100, five gave 300 and one gave 301. Of the studies that reported a maximum threshold to be eligible for inclusion one gave 200, one gave 250 and three gave 299.

## Discussion

Information on COVID-19 disease severity in clinical trials is critical for the interpretation of therapeutic responses. Unfortunately, the summary information reported in the majority of COVID-19 clinical trial publications is insufficient to determine retrospectively, with sufficient accuracy, the distribution of disease severities of the patients included in the studies. In many studies, the key measures of severity were not reported at all. When they were reported they were often incomplete or ambiguous, e.g. being on oxygen-therapy at admission. Many of the

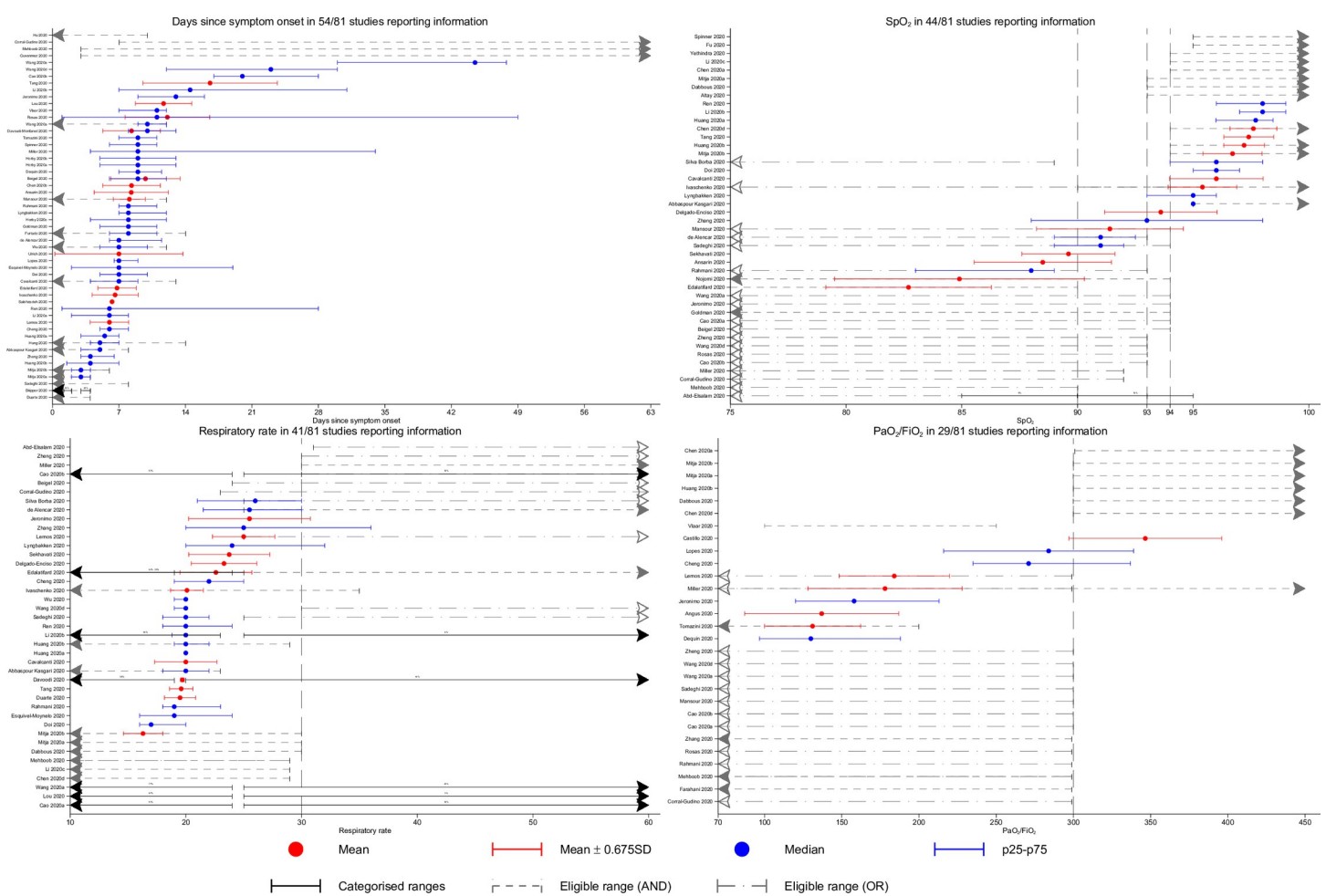

**Fig 5. Available information on days since symptom onset; SpO₂; respiratory rate; PaO₂/FiO₂.** *Dashed vertical lines denote thresholds used by either the NIH; WHO; Covid-NMA or Magic NMA groups to categorise severity.* Eligible range (AND) refers to a criterion that is part of an AND condition and therefore an individual falling into this range would also require at least one other criterion to be met; eligible range (OR) refers to a criterion that is part of an OR condition. The same graph stratified by outpatient composition (all/none/unknown) is presented in S1–S3 Figs. *SpO2: Blood Oxygen Saturation; PaO2/FiO2: Ratio of arterial oxygen partial pressure (PaO2 in mmHg) to fractional inspired oxygen (FiO2 expressed as a fraction)).*

current definitions used for COVID-19 severity combine a mixture of AND and OR logical operators. For example, the WHO guideline definition considers a patient to have severe disease if they have a respiratory rate>30/minute or SpO₂<90% on room air. These are not equivalent. A study that reports proportions of patients above and below these thresholds separately does not provide sufficient information to determine the proportion of patients meeting the criteria for severe infection. In the absence of this information in the trial reports, and with such variability in definitions, guidelines based on summary data are compromised. Analysis of the Individual Participant Data (IPD) is required to assess therapeutic responses in relation to disease severity.

The published literature contains a wide variety of COVID-19 severity threshold criteria, severity definitions and severity categories—many of which are arbitrary (i.e. they have not been calibrated either by mortality or complications). The WHO panel noted that "the oxygen saturation threshold of 90% to define severe covid-19 was arbitrary" [6]. The other indicator thresholds such as respiratory rate or PaO₂/FiO₂ are also arbitrary, and they are also not generally agreed upon. Providing two alternative criteria for severity is also potentially misleading.

Although few would disagree that acute onset of hypoxia resulting in an oxygen saturation of <90% in a patient with previously normal lung function is a sign of severity, a rapid respiratory rate can reflect a number of processes including anxiety. To add to the confusion, during the course of the pandemic some of the definitions were changed e.g. the WHO clinical guidance for COVID-19 published on 27 May 2020 (version 3) defined severity of COVID-19 by clinical indicators, but modified the oxygen saturation threshold from 94% to 90% [14], in order to align with previous WHO guidance [15]. The current severity definitions used by the US-NIH and WHO are expert-based consensus definitions. A measure of severity should be based on available data and outcomes, and include disease specific factors linked to patient prognosis (independent of underlying patient factors). The large variation in mortality observed between trials supposedly including patients with similar levels of disease severity illustrates the problem. While this study did not specifically review the compliance to clinical trial reporting guidelines, several did not follow standard practices. Despite the urgency of reporting clinical outcomes during a pandemic, using CONSORT and other reporting guidelines should remain the standard and be enforced by Editors. In summary, the majority of the RCTs which formed the evidence base for the WHO therapeutic guidelines included mixed populations, with unclear distributions of severity and associated outcomes, and significant variability in observed mortalities even among groups classified as having similar severities.

This is a significant concern because in COVID-19 there is strong evidence for heterogeneity of treatment effects according to disease severity at the time of treatment, as seen with corticosteroids [2,5,7,8]. Fortunately, the large randomised controlled trials in hospitalised patients have provided robust evidence and thus guidance for the management of severe COVID-19, but substantial uncertainty remains for chemoprevention and the treatment of early COVID-19. Epidemiological and therapeutic assessments would benefit substantially from agreement on definitions of severity and full reporting of key clinical measures, so that the quality and thus value of meta-analyses can be improved.

The slow global roll-out of vaccines and the threat of new variants means that effective therapeutics are still needed urgently. Most therapeutic trials on COVID-19 have been on hospitalised patients and most trials reported to date contain insufficient information to classify accurately the range of disease severity at randomisation. Researchers and policy makers must be careful not to over-interpret currently available data. In particular, the extrapolation of "lack of benefit" observed in hospitalised severely ill patients on respiratory support to ambulant mildly ill patients is not warranted. Considering the heterogeneity of disease severity definitions reported here and the amalgamation of patient outcomes reported in the literature, individual patient data (IPD) meta-analyses are needed now to guide and improve therapeutic recommendations in COVID-19. To date, effective data sharing initiatives have been successfully established for longitudinal data, such as the ones from the International Severe Acute Respiratory Infections Consortium (ISARIC) and the Virus Covid-19 Registry [16,17]. Sharing clinical trial data is still, two years after the start of the pandemic, in its infancy. Some funders are making data sharing a requirement attached to their grants, but they rarely enforce it, and the timeline to do so is often left to investigators. In a recently published review of COVID-19 clinical trial registries, intention to share individual participant data was reported in a minority of studies (348/1314, 26%) [18]. Data sharing will only happen if both incentives for investigators and clear policies from funders, scientific journals, regulators, policymakers and other stakeholders are aligned. This paradigm shift is likely to be successful if the research community embraces the culture of "Open Science". Even during a pandemic and its urgent need for openness, this work simply illustrated that open science should be a standard but remains to date a wishful policy. Conducting IPD meta-analyses on clinical trial data require international collaborations to ensure equitable data sharing, following FAIR principles (Fig 6). IPD meta-

## A. CURRENT APPROACH

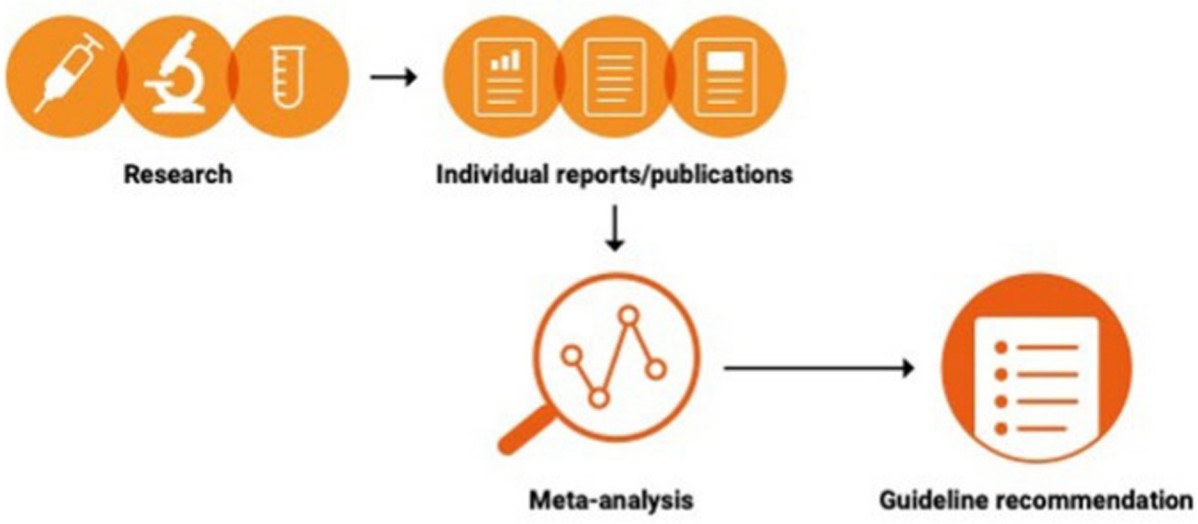

## B. PROPOSED STRATEGY

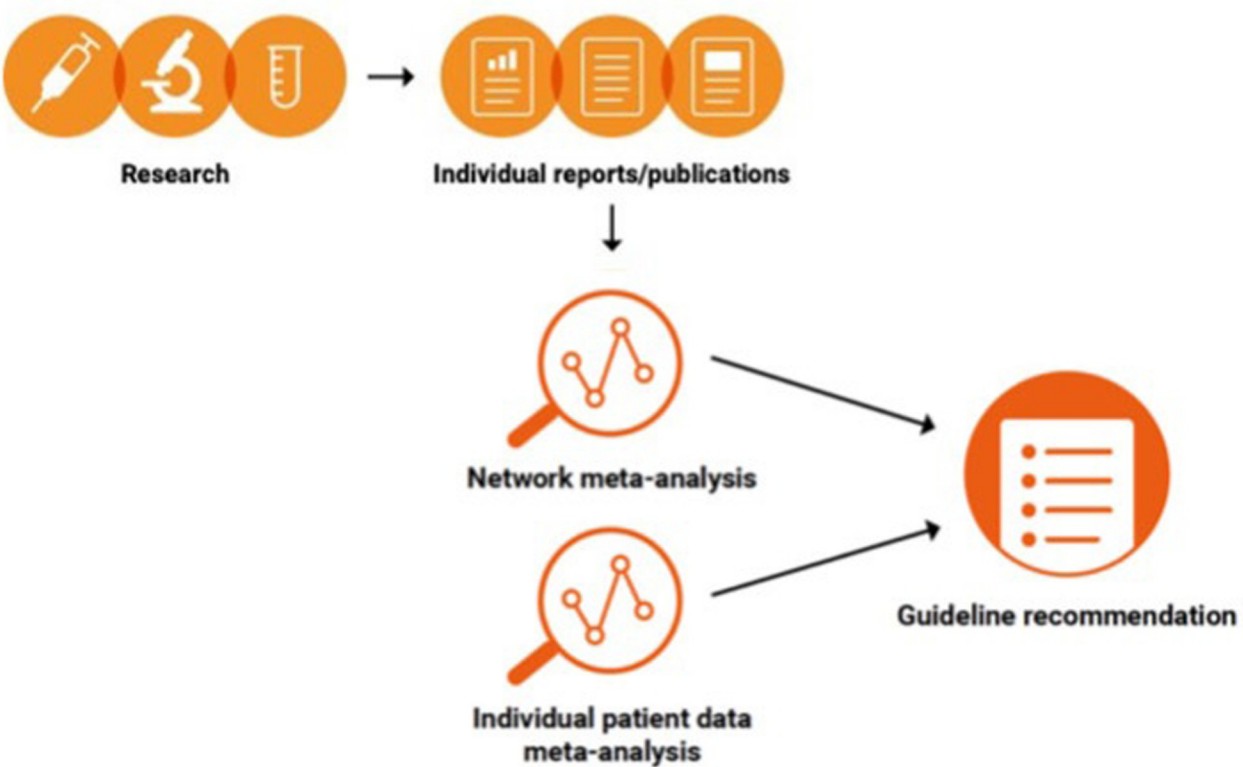

**Fig 6. Proposed strategy to generate COVID-19 treatment guidelines.**

analyses are particularly important for assessing drug efficacy in the large majority of patients who do not yet require hospitalisation.

## Supporting information

**S1 Fig. Available information among studies that only contain outpatients on days since symptom onset; SpO$_2$; respiratory rate; PaO$_2$/FiO$_2$.** *Dashed vertical lines denote thresholds used by either the NIH; WHO; Covid-NMA or Magic NMA groups to categorise severity.* Eligible range (AND) refers to a criterion that is part of an AND condition and therefore an individual falling into this range would also require at least one other criterion to be met; eligible range (OR) refers to a criterion that is part of an OR condition. *SpO2: Blood Oxygen Saturation; PaO2/FiO2: ratio of arterial oxygen partial pressure (PaO2 in mmHg) to fractional inspired oxygen (FiO2 expressed as a fraction)).*
(TIFF)

**S2 Fig. Available information among studies that contain no outpatients on days since symptom onset; SpO$_2$; respiratory rate; PaO$_2$/FiO$_2$.** *Dashed vertical lines denote thresholds used by either the NIH; WHO; Covid-NMA or Magic NMA groups to categorise severity.* Eligible range (AND) refers to a criterion that is part of an AND condition and therefore an individual falling into this range would also require at least one other criterion to be met; eligible range (OR) refers to a criterion that is part of an OR condition. *SpO2: Blood Oxygen Saturation; PaO2/FiO2: ratio of arterial oxygen partial pressure (PaO2 in mmHg) to fractional inspired oxygen (FiO2 expressed as a fraction)).*
(TIFF)

**S3 Fig. Available information among studies with unknown outpatient composition on days since symptom onset; SpO$_2$; respiratory rate; PaO$_2$/FiO$_2$.** *Dashed vertical lines denote thresholds used by either the NIH; WHO; Covid-NMA or Magic NMA groups to categorise severity.* Eligible range (AND) refers to a criterion that is part of an AND condition and therefore an individual falling into this range would also require at least one other criterion to be met; eligible range (OR) refers to a criterion that is part of an OR condition. *SpO2: Blood Oxygen Saturation; PaO2/FiO2: ratio of arterial oxygen partial pressure (PaO2 in mmHg) to fractional inspired oxygen (FiO2 expressed as a fraction)).*
(TIFF)

**S1 Table. Severity definitions.**
(DOCX)

**S2 Table. Trial publication details.**
(DOCX)

**S3 Table. Agreement between WHO—COVID-19 Living Network Meta-Analysis and COVID-NMA initiative groups with respect to possible minimum and maximum severity of trial participants (n = 70).**
(DOCX)

## Acknowledgments

We thank Brittany Maguire for support on the methodological approach, Emile Guerin for his technical support on figures and Lucy Peers for graphic design. This work is commissioned by the COVID-19 clinical trial coalition (https://covid19crc.org/).

## Author Contributions

**Conceptualization:** Philippe J. Guérin, Alistair R. D. McLean, Nicholas J. White.

**Data curation:** Alistair R. D. McLean, Sumayyah Rashan, AbdulAzeez Lawal.

**Formal analysis:** Alistair R. D. McLean, James A. Watson.

**Funding acquisition:** Philippe J. Guérin, Nathalie Strub-Wourgaft.

**Investigation:** Philippe J. Guérin, Sumayyah Rashan.

**Methodology:** Philippe J. Guérin, James A. Watson, Nathalie Strub-Wourgaft.

**Resources:** Philippe J. Guérin, Sumayyah Rashan, AbdulAzeez Lawal.

**Supervision:** Philippe J. Guérin, Nicholas J. White.

**Validation:** Philippe J. Guérin, Sumayyah Rashan, Nathalie Strub-Wourgaft, Nicholas J. White.

**Visualization:** Philippe J. Guérin, James A. Watson, Nathalie Strub-Wourgaft, Nicholas J. White.

**Writing – original draft:** Philippe J. Guérin, Alistair R. D. McLean.

**Writing – review & editing:** Sumayyah Rashan, AbdulAzeez Lawal, James A. Watson, Nathalie Strub-Wourgaft, Nicholas J. White.

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
