## [Decision Letter · Decision Letter 0]

7 Feb 2022

PGPH-D-21-00411

Definitions matter: heterogeneity of COVID-19 disease severity criteria and incomplete reporting compromise meta-analysis

Dear Dr. Guerin,

Thank you for submitting your manuscript to PLOS Global Public Health. The reviewers have recommended minor revisions. Therefore, we invite you to submit a revised version of the manuscript that addresses the points raised during the review process.

We look forward to receiving your revised manuscript.

Kind regards,

Stephen Kerr, PhD

Academic Editor

Journal Requirements:

1. Please provide separate figure files in .tif or .eps format only.  Please ensure that all files are under our size limit of 20MB.  

For more information about how to convert your figure files please see our guidelines: Once you've converted your files to .tif or .eps, please also make sure that your figures meet our format requirements

2. Please update the completed 'Competing Interests' statement, including any COIs declared by your co-authors. If you have no competing interests to declare, please state "The authors have declared that no competing interests exist". Otherwise please declare all competing interests beginning with the statement "I have read the journal's policy and the authors of this manuscript have the following competing interests:"

3. We have noticed that you have uploaded supporting information but you have not included a list of legends.  Please add a full list of legends for all supporting information files (including figures, table and data files) after the references list. 

4. In the online submission form, you indicated that "All the data used in this study are publicly available and properly cited. However, more guided instruction to get access to the data for transparency and reproducibility will be provided on request.". All PLOS journals now require all data underlying the findings described in their manuscript to be freely available to other researchers, either 1. In a public repository, 2. Within the manuscript itself, or 3. Uploaded as supplementary information.

5. Please amend your detailed Financial Disclosure statement. This is published with the article, therefore should be completed in full sentences and contain the exact wording you wish to be published.

ii). State the initials, alongside each funding source, of each author to receive each grant.

iii). State what role the funders took in the study. If the funders had no role in your study, please state: “The funders had no role in study design, data collection and analysis, decision to publish, or preparation of the manuscript.”

Additional Editor Comments (if provided):

Reviewer 1.

Overall: This is a timely and informative study. The authors carefully reviewed the current treatment guidelines and the studies used to inform the guidelines and provided logical and evidence-based critiques on the inconsistency in definitions/categorizations of COVID 19 disease severity. I (the reviewer) consider myself as a population health researcher with limited clinical experience, but I may be able to provide perspectives that are representative of the potential readers of the journal. Another overall comment is that the figures in this study are very informative, but I am afraid they are not self-explanatory enough. In general, more detailed notes/legends will be very helpful. Please consider the following recommendations.

(I found it very inconvenient to write comments without the availability of line numbers. Please consider adding line numbers if the manuscript needs reviews in the future)

Introduction:

(1) The concepts of “disease severity” and “stage of the disease” have been used interchangeably in the current study. But to me, the stage of the disease seems to be more related to time/phase instead of severity, especially when being used to determine, for example, whether a patient is at the initial infection phase of the disease where the viral burden peaks, or at the later phase when inflammatory processes dominate. Further clarifying these concepts is needed because the introduction section particularly emphasized the importance of determining disease course, and how crucial it is in terms of therapeutic recommendations (the timing of when to administer medicines). In the intro, the authors may consider at least briefly answering why a more consistent and more precise severity assessment can help better identify the stage of the disease, and thus guide and improve therapeutic recommendations.

(2) Though already specified in the main text, from a reader’s perspective, it is still helpful and necessary to have notes explaining the abbreviation used in the figures. (For example, Figure 1, respiratory rate and oxygen saturation level.)

(3) In the last paragraph of the introduction, the aim of the study is phrased “We sought to determine the feasibility of determining retrospectively the disease severity of patients included in the COVID-19 clinical trials upon which the current WHO living therapeutic guidelines are based.” I personally think that this sentence can be potentially reframed and simplified. Also, the authors may reconsider whether the aim of the study is determining the feasibility of the current measures, or examining the consistency or comparability of the current measures.

Materials and Methods:

(4) Under the subtitle of “categorical indicators of disease severity”, since the categories indicate severity, it will be very helpful if they are ordered accordingly, starting from the least severe category to the most severe one.

Results:

(5) Figure 2 needs notes/legends. Please specify the meaning of red dots and green triangles. Please specify the difference between the red and black dash lines.

(6) Figure 3 needs notes/legends and reorganization. Please specify the meaning of the colored dots. Please relocate the letter A and letter B, so they do not overlap with the figure title.

(7) Figure 5 needs notes/legends. Please specify what the different colored arrows mean.

Discussion:

(8) The paper clearly pointed out the drawbacks of the current severity measurement. It might be obvious to the authors that how individual patient data meta-analysis will be able to contribute, but to people who are not very familiar with clinical research, more details on how the proposed solution will provide evidence for a consistent and accurate severity definition will be extremely helpful. In other words, I do think the current research revealed very important issues of the current literature, but I do not think it indicates the future direction of the research clearly enough.

Reviewer 2

This is a cleverly conceived methods paper which highlights the difficulties of both setting up trials for a new disease in the absence of clear understanding of the disease pathogenesis, and how interventions might vary in effectiveness at different disease stages, and making recommendations on therapeutics at a global level. It is useful to highlight how the deficiencies described in the paper have compromised meta-analysis and recommendations, but I would like the authors to expand the discussion on why this has happened, and strategies so that it would not happen again in future pandemics.

The abstract recommendations are a logical conclusion from the findings, but could there be a final line on how this could be achieved?

I think it would be useful to have the individual studies that have been used to generate the meta-analytic findings are listed as references in a supplement. There are no descriptions of what software was used to generate the figures and calculate the summary statistics, so it would be useful to include this information in the methods.

I agree with the authors that standard definitions is a very important initiative, but given that authors have not reported key clinical measures used in their studies, could the authors propose a strategy for achieving this? Poor reporting has been noted for many years, and the equator-network has sought to develop guidelines to improve reporting standards.

Many journals have accepted the manuscripts with incomplete reporting of key clinical measures, even though CONSORT and other reporting guidelines exist, and these guidelines ask for eligibility criteria, completely defined pre-specified primary and secondary outcome measures, including how and when they were assessed. Part of the problem is therefore manuscripts accepting reports, when the ideal information in the reports is not given. For journals that ask require reporting guideline checklists, this responsibility ultimately is a failure of the editorial and review process. The discussion should note these deficiencies, and suggest a strategy for improvement.

I suspect some of this poor reporting reflects the haste of authors to publish their work, and the haste of journals to publish comparatively new information. International collaboration has been poor in many aspects of this pandemic, including investigating viral origins, and ensuring equitable access to vaccines.

I also agree that international collaborations are required to ensure equitable data sharing. Some, but not all journals require that data is available in a public registry, and authors are often reluctant to agree to this demand. Do the authors have suggestions on how this aspect could be improved?

Reviewers' comments:

Reviewer's Responses to Questions

**Comments to the Author**

1. Does this manuscript meet PLOS Global Public Health’s publication criteria? Is the manuscript technically sound, and do the data support the conclusions? The manuscript must describe methodologically and ethically rigorous research with conclusions that are appropriately drawn based on the data presented.

Reviewer #1: Yes

Reviewer #2: Yes

2. Has the statistical analysis been performed appropriately and rigorously?

Reviewer #1: Yes

Reviewer #2: Yes

3. Have the authors made all data underlying the findings in their manuscript fully available (please refer to the Data Availability Statement at the start of the manuscript PDF file)?

Reviewer #1: Yes

Reviewer #2: Yes

4. Is the manuscript presented in an intelligible fashion and written in standard English?

Reviewer #1: Yes

Reviewer #2: Yes

5. Review Comments to the Author

Reviewer #1: Overall: This is a timely and informative study. The authors carefully reviewed the current treatment guidelines and the studies used to inform the guidelines and provided logical and evidence-based critiques on the inconsistency in definitions/categorizations of COVID 19 disease severity. I (the reviewer) consider myself as a population health researcher with limited clinical experience, but I may be able to provide perspectives that are representative of the potential readers of the journal. Another overall comment is that the figures in this study are very informative, but I am afraid they are not self-explanatory enough. In general, more detailed notes/legends will be very helpful. Please consider the following recommendations.

(I found it very inconvenient to write comments without the availability of line numbers. Please consider adding line numbers if the manuscript needs reviews in the future)

Introduction:

(1) The concepts of “disease severity” and “stage of the disease” have been used interchangeably in the current study. But to me, the stage of the disease seems to be more related to time/phase instead of severity, especially when being used to determine, for example, whether a patient is at the initial infection phase of the disease where the viral burden peaks, or at the later phase when inflammatory processes dominate. Further clarifying these concepts is needed because the introduction section particularly emphasized the importance of determining disease course, and how crucial it is in terms of therapeutic recommendations (the timing of when to administer medicines). In the intro, the authors may consider at least briefly answering why a more consistent and more precise severity assessment can help better identify the stage of the disease, and thus guide and improve therapeutic recommendations.

(2) Though already specified in the main text, from a reader’s perspective, it is still helpful and necessary to have notes explaining the abbreviation used in the figures. (For example, Figure 1, respiratory rate and oxygen saturation level.)

(3) In the last paragraph of the introduction, the aim of the study is phrased “We sought to determine the feasibility of determining retrospectively the disease severity of patients included in the COVID-19 clinical trials upon which the current WHO living therapeutic guidelines are based.” I personally think that this sentence can be potentially reframed and simplified. Also, the authors may reconsider whether the aim of the study is determining the feasibility of the current measures, or examining the consistency or comparability of the current measures.

Materials and Methods:

(4) Under the subtitle of “categorical indicators of disease severity”, since the categories indicate severity, it will be very helpful if they are ordered accordingly, starting from the least severe category to the most severe one.

Results:

(5) Figure 2 needs notes/legends. Please specify the meaning of red dots and green triangles. Please specify the difference between the red and black dash lines.

(6) Figure 3 needs notes/legends and reorganization. Please specify the meaning of the colored dots. Please relocate the letter A and letter B, so they do not overlap with the figure title.

(7) Figure 5 needs notes/legends. Please specify what the different colored arrows mean.

Discussion:

(8) The paper clearly pointed out the drawbacks of the current severity measurement. It might be obvious to the authors that how individual patient data meta-analysis will be able to contribute, but to people who are not very familiar with clinical research, more details on how the proposed solution will provide evidence for a consistent and accurate severity definition will be extremely helpful. In other words, I do think the current research revealed very important issues of the current literature, but I do not think it indicates the future direction of the research clearly enough.

Reviewer #2: This is a cleverly conceived methods paper which highlights the difficulties of both setting up trials for a new disease in the absence of clear understanding of the disease pathogenesis, and how interventions might vary in effectiveness at different disease stages, and making recommendations on therapeutics at a global level. It is useful to highlight how the deficiencies described in the paper have compromised meta-analysis and recommendations, but I would like the authors to expand the discussion on why this has happened, and strategies so that it would not happen again in future pandemics.

The abstract recommendations are a logical conclusion from the findings, but could there be a final line on how this could be achieved?

I think it would be useful to have the individual studies that have been used to generate the meta-analytic findings are listed as references in a supplement. There are no descriptions of what software was used to generate the figures and calculate the summary statistics, so it would be useful to include this information in the methods.

I agree with the authors that standard definitions is a very important initiative, but given that authors have not reported key clinical measures used in their studies, could the authors propose a strategy for achieving this? Poor reporting has been noted for many years, and the equator-network has sought to develop guidelines to improve reporting standards.

Many journals have accepted the manuscripts with incomplete reporting of key clinical measures, even though CONSORT and other reporting guidelines exist, and these guidelines ask for eligibility criteria, completely defined pre-specified primary and secondary outcome measures, including how and when they were assessed. Part of the problem is therefore manuscripts accepting reports, when the ideal information in the reports is not given. For journals that ask require reporting guideline checklists, this responsibility ultimately is a failure of the editorial and review process. The discussion should note these deficiencies, and suggest a strategy for improvement.

I suspect some of this poor reporting reflects the haste of authors to publish their work, and the haste of journals to publish comparatively new information. International collaboration has been poor in many aspects of this pandemic, including investigating viral origins, and ensuring equitable access to vaccines.

I also agree that international collaborations are required to ensure equitable data sharing. Some, but not all journals require that data is available in a public registry, and authors are often reluctant to agree to this demand. Do the authors have suggestions on how this aspect could be improved?

6. PLOS authors have the option to publish the peer review history of their article (what does this mean?). If published, this will include your full peer review and any attached files.

**Do you want your identity to be public for this peer review?** For information about this choice, including consent withdrawal, please see our Privacy Policy.

Reviewer #1: No

Reviewer #2: No

---

## [Decision Letter · Decision Letter 1]

8 Jun 2022

Definitions matter: heterogeneity of COVID-19 disease severity criteria and incomplete reporting compromise meta-analysis

PGPH-D-21-00411R1

Dear Prof Guerin,

We are pleased to inform you that your manuscript 'Definitions matter: heterogeneity of COVID-19 disease severity criteria and incomplete reporting compromise meta-analysis' has been provisionally accepted for publication in PLOS Global Public Health.

Best regards,

Stephen Kerr

Academic Editor

Reviewer Comments (if any, and for reference):

Reviewer's Responses to Questions

**Comments to the Author**

1. If the authors have adequately addressed your comments raised in a previous round of review and you feel that this manuscript is now acceptable for publication, you may indicate that here to bypass the “Comments to the Author” section, enter your conflict of interest statement in the “Confidential to Editor” section, and submit your "Accept" recommendation.

Reviewer #2: All comments have been addressed

2. Does this manuscript meet PLOS Global Public Health’s publication criteria? Is the manuscript technically sound, and do the data support the conclusions? The manuscript must describe methodologically and ethically rigorous research with conclusions that are appropriately drawn based on the data presented.

Reviewer #2: Yes

3. Has the statistical analysis been performed appropriately and rigorously?

Reviewer #2: Yes

4. Have the authors made all data underlying the findings in their manuscript fully available (please refer to the Data Availability Statement at the start of the manuscript PDF file)?

Reviewer #2: Yes

5. Is the manuscript presented in an intelligible fashion and written in standard English?

Reviewer #2: Yes

6. Review Comments to the Author

Reviewer #2: Thank you for responding to my comments. This is an important contribution to the scientific literature

7. PLOS authors have the option to publish the peer review history of their article (what does this mean?). If published, this will include your full peer review and any attached files.

**Do you want your identity to be public for this peer review?** For information about this choice, including consent withdrawal, please see our Privacy Policy.

Reviewer #2: No
